

# Complexity of frustration: A new source
# of non-local non-stabilizerness

**Jovan Odavić[1*], Tobias Haug[2†], Gianpaolo Torre[1‡], Alioscia Hamma[3,4,5°],
Fabio Franchini[1§] and Salvatore Marco Giampaolo[1¶]**

**1** Ruđer Bošković Institute, Bijenička cesta 54, 10000 Zagreb, Croatia
**2** QOLS, Blackett Laboratory, Imperial College London SW7 2AZ, UK
**3** Dipartimento di Fisica Ettore Pancini, Università degli Studi di Napoli Federico II,
Via Cinthia, 80126 Fuorigrotta, Napoli NA
**4** Physics Department, University of Massachusetts Boston, 02125, USA
**5** INFN, Sezione di Napoli, Italy

★ jodavic@irb.hr , † thaug@ic.ac.uk , ‡ gianpaolo.torre@irb.hr ,
○ alioscia.hamma@unina.it , § fabio.franchini@irb.hr , ¶ sgiampa@irb.h

## Abstract

We advance the characterization of complexity in quantum many-body systems by examining $W$-states embedded in a spin chain. Such states show an amount of non-stabilizerness or *"magic"*, measured as the Stabilizer Rényi Entropy, that grows logarithmically with the number of qubits/spins. We focus on systems whose Hamiltonian admits a classical point with extensive degeneracy. Near these points, a Clifford circuit can convert the ground state into a $W$-state, while in the rest of the phase to which the classical point belongs, it is dressed with local quantum correlations. Topological frustrated quantum spin-chains host phases with the desired phenomenology, and we show that their ground state's Stabilizer Rényi Entropy is the sum of that of the $W$-states plus an extensive local contribution. Our work reveals that $W$-states/frustrated ground states display a non-local degree of complexity that can be harvested as a quantum resource and has no counterpart in GHZ states/non-frustrated systems.



# 1  Introduction

The problem of simulating quantum states is, generally, intractable for classical computers. For this reason, Feynman put forward the notion of a quantum computer [1] as only a quantum device would be able to simulate a generic quantum system efficiently. This necessity is of particular relevance for states of quantum many-body systems that can be used to accommodate new exotic phenomena of matter like quantum criticality [2], topological order [3–5], exotic metals without quasi-particle excitations [6,7] quantum systems away from equilibrium like systems of ultra-cold atomic gases [8–10], etc.

However, the picture that has emerged after years of research is much more multifaceted than what Feynman weighed. Indeed, the large amounts of studies of quantum properties, most notably entanglement, in quantum many-body systems have fostered and given an impulse towards relevant progress in their simulation. It is now well-known that states defined in one-dimensional systems obeying area law [11,12] can be efficiently represented and manipulated using matrix-product states (MPS) techniques [13,14], tensor-networks [15,16], entanglement renormalization schemes [17] and other computational procedures. Hence, the complexity of quantum simulations does not affect all quantum states and does not arise from entanglement alone. Certain classes of states can be, at the same time, highly entangled and efficiently simulated on a classical computer.

This is the case of the stabilizer states [18], namely those states that can be obtained from the computational basis using both Pauli operations and the so-called Clifford gates. The Clifford gates are a set of three quantum gates, namely the Hadamard, CNOT, and the Phase gate [19,20], that are very efficient in creating entanglement [21,22] but do not provide any quantum supremacy [1,23–25]. Since they can be efficiently represented (i.e. with a cost that increases only polynomially with the size of the system) on a classical computer, there is no information processing that a quantum computer could do by Clifford resources that would not be efficiently performed by a classical computer. On the other hand, as soon as such circuits get doped with non-Clifford resources, their entanglement pattern becomes more complex, driving a transition to universality and quantum chaos [26–29].

In the context of quantum many-body systems, the study of non-stabilizerness has been limited, probably because of the hardness of computing the existing measures for this resource, which has been dubbed *magic* in the folklore [30–33]. Among the few exceptions [34–36] one that is worth noting is [37] where the authors show the usefulness of quantum many-body states with non-Clifford resources for quantum computation.

Lately, however, a new measure of non-stabilizerness has been introduced [38] as the Stabilizer Rényi Entropy (SRE), which can be computed efficiently for MPSs [39] but is also amenable to experimental measurement [40,41]. In Ref. [42], by exploiting the computability of the SRE, it was shown that the ground state of an Ising spin chain in a transverse field, despite obeying entanglement area law [11], does possess an extensive amount of non-stabilizerness. In the gapped phases, the SRE can be resolved by local quantities, i.e. its value can be well approximated as the sum of the SRE of its parts. This picture fails at the quantum critical point, where the correlation length diverges [43,44], and the entanglement shows a logarithmic violation of the area law. Here, the local approximation fails with a large offset error due to entanglement and magic getting a diverging logarithmic correction [39] which cannot be captured by local measures. Therefore, one has to consider very large blocks to get a reasonable approximation of the exact result. It is worth reminding that at critical points decimation schemes like the MPS encounter a hurdle, because of the logarithmic divergence of entanglement. The resulting picture is of a quantum complexity emerging from a delocalization of non-stabilizerness, i.e., the impossibility of resolving SRE in terms of local quantities.

In this paper, we explore the delocalization of SRE in quantum many-body systems, by considering a class of states – the $W$-states [45] – which are a global superposition of a macroscopic number of factorized states. By embedding these states as the ground states of a quantum spin chain and adding additional interactions to dress these states with local correlation, we achieve a detailed characterization of various contributions to their SRE.

Notice that, although the results presented in [42] are associated with a particular model, the picture there obtained is expected to be quite general. Indeed, these results agree with the ones expected for ground states obtained by exploiting quasi-adiabatic continuation [46] from a classical point with a finite degeneracy. In this context, a classical point is meant as a point in the phase diagram where the Hamiltonian reduces to a sum of terms that commutes with each other and the SRE of the symmetric mixture of all its elements vanishes. The deformations of the Hamiltonian associated with the quasi-adiabatic continuation, induce the growth of local quantum correlations, which explains both the extensivity and the local nature of the SRE [47].

In this work, we consider a different situation, where quantum many-body systems admit classical points with ground state manifolds whose number of elements scale with $L$. These states can be grouped into a finite number of families, each of which admits a base made of states that can be obtained from each other by spatial translations. To provide an example, this is the case of the spin one-half chains with topological frustration, which boils down to the frustrated Ising model at the classical point [48–50]. This last system admits a ground state manifold whose basis is the set of single kink states, i.e. states with perfect Néel order except for a ferromagnetic defect (with two parallelly aligned neighboring spins) which constitute a domain wall switching between Néel orders. The $2L$ kink states can be arranged in 2 different families of states (with different parities for the magnetization), in which each element can be obtained from the others by spatial translations (i.e., by shifting the kink).

Moving away from the classical point, generally, the extensive degeneracy gets lifted, and, depending on the symmetry of the term competing with the Ising one, the ground state can be represented by a symmetric coherent superposition of the kink states. This happens, among other cases, when the competing term commutes with the parity of the magnetization along a direction orthogonal to that of the Ising interaction [51–53]. The symmetric linear superposition of the kink states can be connected, through a Clifford circuit, with the $W$-states, whose SRE will be shown to grow logarithmically in $L$. This non-vanishing SRE of the $W$-states cannot be resolved in terms of local quantities, since it comes from the delocalized nature of the linear superposition. Therefore, in the proximity of the classical point, we expect an SRE that, instead of vanishing as in [42], depends logarithmically on $L$, thus signaling that they belong to a different and inequivalent class of states, compared to the usual case. Moving further away from the classical point, a finite correlation length is developed which adds to this logarithmic growth part and an extensive correction, similar in nature to the one discussed above. In fact, in the case of topologically frustrated systems, this second term can be traced back to the SRE of the corresponding unfrustrated system.

The manuscript is organized as follows. First, in Sect. 2 we introduce the $W$-states, and we evaluate analytically their SRE and its dependence on the system size. Next, in Sect. 3 we show how the SRE of the $W$-states is related to the symmetric linear superposition of magnetic defects (kinks) states which, in the proximity of a classical point, well-describes the ground state of a family of topologically frustrated one-dimensional spin-1/2 models. Next, in Sect. 4 we analyze in detail some examples of topologically frustrated integrable models, and we underline differences and analogies with the unfrustrated counterparts. Finally, in Sect. 5 we draw our conclusion.

## 2  W-states

Let us start by recalling that, to quantify the amount of non-stabilizerness for a generic state defined on a one-dimensional system made of $L$ qubits, it is possible to use the Stabilizer 2-Rényi Entropy (SRE) [38] that is defined as

$$\mathcal{M}_2(|\psi\rangle) = -\log_2\left(\frac{1}{2^L}\sum_P \langle\psi|P|\psi\rangle^4\right), \tag{1}$$

where the sum on the right-hand side runs over all possible Pauli strings $P = \bigotimes_{j=1}^{L} P_j$ for $P_j \in \{\sigma_j^0, \sigma_j^x, \sigma_j^y, \sigma_j^z\}$ where $\sigma_j^0$ stands for the identity operator on the $j$-th qubit.

Let us start by considering a set of $L$ states, each of which is an element of the computational basis. Defining $T$ the translation operator, we assume that each element $|\psi_j\rangle$ satisfies the following relation $|\psi_{j+1}\rangle = T|\psi_j\rangle$. Since each $|\psi_j\rangle$ is an element of the computational basis, it is an eigenstate of $2^L$ Pauli strings with associated eigenvalues equal to $\pm 1$. It has a vanishing expectation value for all the other Pauli strings. Therefore, its SRE vanishes identically as well as the one of a classical state. However, if instead of considering only one of the elements of the set, we take into account the translational invariant linear combination of all of them, i.e. the state $|\chi\rangle = L^{-1/2}\sum_j |\psi_j\rangle$ we obtain an SRE different from zero. To evaluate the SRE of $|\chi\rangle$, let us observe that regardless of the particular expression of $|\psi_j\rangle$, since it is an eigenstate of a Pauli string, it is always possible, by exploiting only local rotations, to map it into the state $|j\rangle = \sigma_j^z|-\rangle^{\otimes L}$ where $|\pm\rangle$ stand for the eigenstates of $\sigma^x$ with eigenvalues $\pm 1$. Hence, the symmetric superposition $|\chi\rangle$ is equivalent to the well-known $W$-states [45] defined as

$$|W\rangle = \frac{1}{\sqrt{L}}\sum_{j=1}^{L}\sigma_j^z|-\rangle^{\otimes L}. \tag{2}$$

$W$-states play a key role in the theory of quantum information since they maximize the multipartite entanglement [45] while retaining the maximum amount of bipartite entanglement after local measurement on one of its part [54]. As such, they are considered good candidates for realizing quantum memories [55].

From the expression of $|W\rangle$ in (2) it is possible to evaluate analytically the value of $\mathcal{M}_2(|W\rangle)$ which turns out to be equal to

$$\mathcal{M}_2(|W\rangle) = 3\log_2(L) - \log_2(7L - 6) \tag{3}$$

(see Appendix A for detailed proof).

The result in (3) shows that there is a whole class of states, that is the $W$-states and all other states that can be obtained from it with the help of a stabilizer circuit, whose SRE displays a logarithmic dependence on the system size. This class is different from the one obtained by using Clifford circuits on a fully separable state, a class to which also the GHZ states [56] belongs and which is characterized by a zero non-stabilizerness.

## 3  Non-stabilizerness close to a classical point

Let us now turn back to consider a quantum many-body problem. If one of the states of the set $\{|\psi_j\rangle\}$ is a ground state of a translationally-invariant Hamiltonian at its classical point, then all the elements of the set, as well as all the possible linear combinations of them, are also ground states of the Hamiltonian. This implies that the Hamiltonian holds, at its classical

point, an extensive degeneracy. An example of a situation like the one described can be found, for instance, in classical (non-disordered) frustrated systems [57–60].

By turning on a quantum (competing) interaction, in general, the massive degeneracy is, at least partially, lifted. In agreement with perturbation theory, the ground state can be well approximated by a linear combination of the elements in the set $\{|\psi_j\rangle\}$. It is not possible to make a general statement about the particular linear combination that minimizes the energy, since it strongly depends on both the expression of the classical Hamiltonian and on the nature of the perturbation. To have a taste of the richness of this phenomenology, see Ref. [61] for the case of one-dimensional topologically frustrated models. Therefore, we are forced to choose a particular family of models.

From now on, we focus on the translationally invariant one-dimensional model whose Hamiltonian can be written in the form

$$H = J \sum_{j=1}^{L} \sigma_j^x \sigma_{j+1}^x - \lambda \sum_{j=1}^{L} O_j. \tag{4}$$

The parameter $\lambda$ allows for tuning the relative weight of the two terms. At $\lambda = 0$ (classical point), the Hamiltonian boils down to a simple 1D Ising model that, in the presence of topological frustration, shows the extensive degeneracy in the ground state manifold that we are looking for. In a translationally invariant 1D system, topological frustration is induced by choosing antiferromagnetic interactions ($J = 1$) and by enforcing the so-called *frustrated boundary conditions* (FBC) that are imposed by setting: 1) periodic boundary conditions ($\sigma_j^\alpha = \sigma_{j+L}^\alpha \ \forall j, \alpha$); 2) odd number of sites ($L = 2M + 1$ for any strictly positive integer $M$).

The family of Hamiltonians in (4) features a second term in competition with the Ising one which preserves translational invariance, violates the parity symmetry along the direction of the Ising interaction but preserves the one with respect to an orthogonal direction, in our case the $z$ direction. Moreover, to exploit the methods presented in Ref. [42], we also require the resulting model to be mappable to a free fermionic one, by making use of the Jordan-Wigner transformation [44]. This family of Hamiltonians is extremely wide and includes, among others, the transverse field Ising model (TFIM) [43], obtained by setting $O_j = \sigma_j^z$ and the Cluster-Ising model (CIM) [62, 63], that is the simplest example of the family of Cluster models [64, 65], which is obtained choosing $O_j = \sigma_{j-1}^y \sigma_j^z \sigma_{j+1}^y$.

At the classical point $\lambda = 0$, the Hamiltonian in (4) admits a ground state manifold with a degeneracy equal to $2L$, as an effect of Kramer's degeneracy theorem. This manifold is described by the union of two extensive sets of states, which are $\{|k\rangle = T^{k-1} \bigotimes_{j=1}^{M} \sigma_{2j}^z |-\rangle^{\otimes L}\}$ and $\{|k'\rangle = T^{k-1} \bigotimes_{j=1}^{M} \sigma_{2j}^z |+\rangle^{\otimes L}\}$ for all $k$ and $k'$ running from 1 to $L = 2M + 1$ where $|\pm\rangle$ are the eigenstates of $\sigma^x$ with eigenvalues $\pm 1$. The elements of these two sets of states are known as kink states or domain-wall states and are Néel's states with a localized magnetic defect (two neighboring spins parallelly oriented which interpolate between the two Néel orders) [61].

Depending on the choice of $O_j$, a finite $\lambda$ reduces the extensive ground state degeneracy of the classical point to a finite odd number. In the cases analyzed in this paper, we have two different behaviors. For the topologically frustrated TFIM, we have that the degeneracy is completely removed and the dimension of the ground state manifold is always equal to 1 [51, 52]. On the contrary, for the CIM, we have two different situations. If $L$ is odd, but it is not an integer multiple of 3 the degeneracy is completely lifted, and we obtain a ground state manifold whose dimension is equal to 1. On the other hand, if $L$ is simultaneously odd and an integer multiple of 3 the dimension of the ground state manifold is equal to 3. [53]. However, independently of the degree of the ground state degeneracy, as the lowest energy state we always find the symmetric superposition of kink states with zero momentum that we can write as

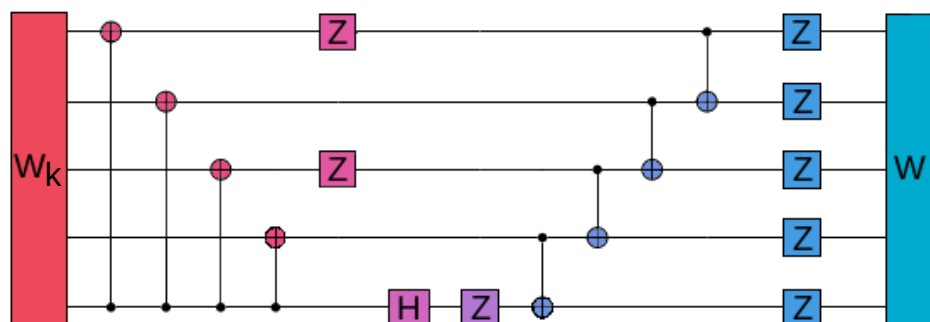

Figure 1: Pictorial representation of the Clifford circuit $\hat{\mathcal{S}}$ in (6) for $L = 5$. The **H** and **Z** boxes stand respectively for the *Hadamard computational* and the $\sigma^z$ operator on the chosen qubit. The CNOT gates involve two qubits and are represented by a line connecting a black dot, indicating the qubit that acts as the controller, and a colored circle signaling the one that can be flipped.

$$|W_k\rangle = \frac{1}{\sqrt{2L}} \sum_{k=1}^{L} (|k\rangle + |k'\rangle). \tag{5}$$

The $|W_k\rangle$ looks similar to $|W\rangle$ and can be obtained from it using a simple stabilizer circuit. This means that we can write $|W_k\rangle = \hat{\mathcal{S}}|W\rangle$ where $\hat{\mathcal{S}}$ is the stabilizer circuit

$$\hat{\mathcal{S}} = \prod_{j=1}^{L-1} \mathbf{C}(L, L-j) \left( \prod_{j=1}^{M} \sigma_{2j-1}^z \right) \mathbf{H}(L) \sigma_L^z \prod_{j=1}^{L-1} \mathbf{C}(j, j+1) \Pi^z. \tag{6}$$

In (6) $\mathbf{H}(j) = \frac{1}{\sqrt{2}}(\sigma_j^x + \sigma_j^z)$ stands for the *Hadamard gate* acting on the $j$-th qubit, while $\mathbf{C}(j, l) = \exp\left[ \iota \frac{\pi}{4}(1 - \sigma_j^x)(1 - \sigma_l^z) \right]$ is the *CNOT gate* on the $l$-th qubit controlled by the value of the $j$-th one while $\Pi^z = \bigotimes_{j=1}^{L} \sigma_j^z$ is the parity operator along $z$. It is worth underlining that since the operators in the second tensor product of the CNOT operators do not commute each other, the product must read as $\prod_{j=1}^{L-1} \mathbf{C}(j, j+1) = \mathbf{C}(L-1, L) \cdot \mathbf{C}(L-2, L-1) \cdot \ldots \cdot \mathbf{C}(1, 2)$. This circuit is depicted in Fig. 1 in the case of $L = 5$. Since $\mathcal{M}_2$ is invariant under stabilizer Clifford circuits, we have $\mathcal{M}_2(|W_K\rangle) = \mathcal{M}_2(|W\rangle)$. Thus, for systems that satisfy our hypothesis, in the proximity of the classical point, the SRE does not vanish but scales logarithmically as in eq. (3).

## 4 Analysis of some topologically frustrated models

Moving further away from the classical point with extensive degeneracy, the approximation we used in the previous section ceases to be valid. Therefore, We must determine the exact expression for the ground state and obtain the value of the SRE from it. This procedure cannot be performed with general arguments such as those used up to now, and the problem must be analyzed case by case. In this paper, we will focus on two different models, namely the topologically frustrated version of both the TFIM and the CIM, whose ground states have been analyzed in previous works [51, 53] exploiting the Jordan-Wigner transformations that allow mapping the spin systems in free-fermionic ones.

In Fig. 2, we depict the results obtained for the SRE as a function of $\lambda$ for both models. For the sake of clarity, in the case of CIM, we split the discussion in two, depending on whether the size of the system is or is not an integer multiple of 3. This choice is justified not only from the presence or absence of a ground state degeneracy in the frustrated case but also from the fact

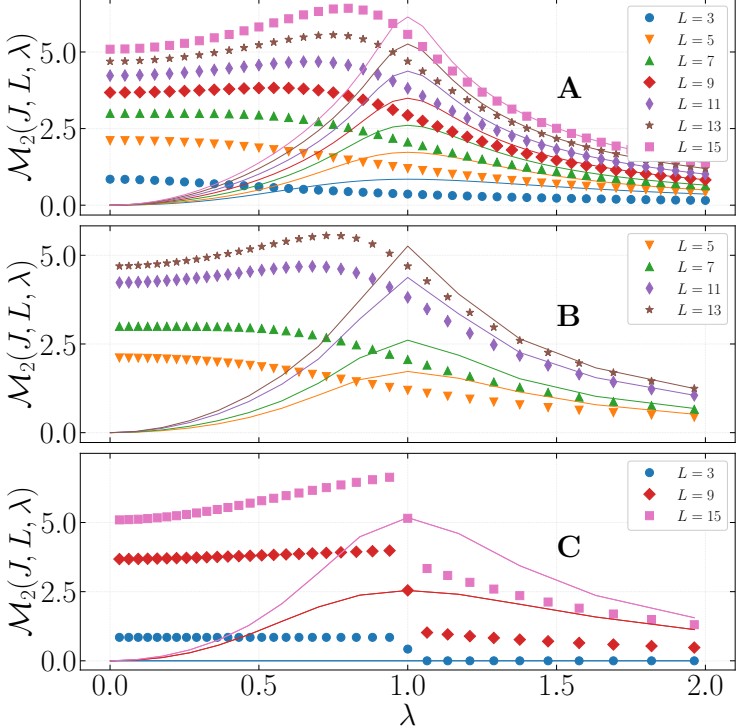

Figure 2: Behavior of the SRE as a function of the parameter $\lambda$ for the TFIM (Panel A), CIM with odd $L$ not integer multiple of 3 (Panel B) and CIM with $L$ odd and integer multiple of 3 (Panel C). Both models show a critical point at $\lambda = 1$ where the unfrustrated models reach a local maximum. In all panels, the points represent the value of SRE obtained for frustrated systems ($J = 1$) while the lines depict the SRE for unfrustrated ones ($J = -1$).

that even the non-frustrated models exhibit two different behaviors for $L$ equal or different from an integer multiple of 3 [62].

In all panels, we see similar behaviors. For the unfrustrated models, the SRE vanishes at the classical point, increases with $\lambda$ reaching the maximum at the critical value $\lambda = 1$, and then decreases vanishing in the limit of diverging $\lambda$. The only exception to this picture is the CIM with $L = 3$ which clearly shows a pathological behavior due to the fact that the cluster interaction extends to the whole system. On the contrary, the SRE for the frustrated model always starts with a non-zero value, as predicted in eq. (3), then, depending on the size of the system, can or cannot reach a maximum *before* the quantum critical point. Increasing $L$ such a maximum becomes both more evident and closer to the phase transition. Above the critical point, the behaviors of the SRE for the frustrated systems are similar to the unfrustrated ones and tend to coincide as $\lambda$ increases.

On the other hand, in Fig. 3, for several values of $\lambda$, we evaluate the SRE as a function of $L$ for the ground states of the unfrustrated models (left column) and the frustrated ones (right column). In agreement with the results obtained in [42], regardless of the value of $\lambda$, the SRE for the unfrustrated models always displays a linear dependence on $L$. Instead, the picture of the frustrated models is much richer. The linear dependence on $L$ is preserved for both the critical ($\lambda = 1$), and the gapped case ($\lambda > 1$). On the contrary, for $\lambda < 1$ we see a non-linear dependence on $L$.

While it is easy to extrapolate the behavior of the SRE for the unfrustrated systems with large $L$, thanks to the linear trend visible even at small sizes, for the frustrated models the situation is more complex. Analyzing the data, we can identify two clearly different behaviors.

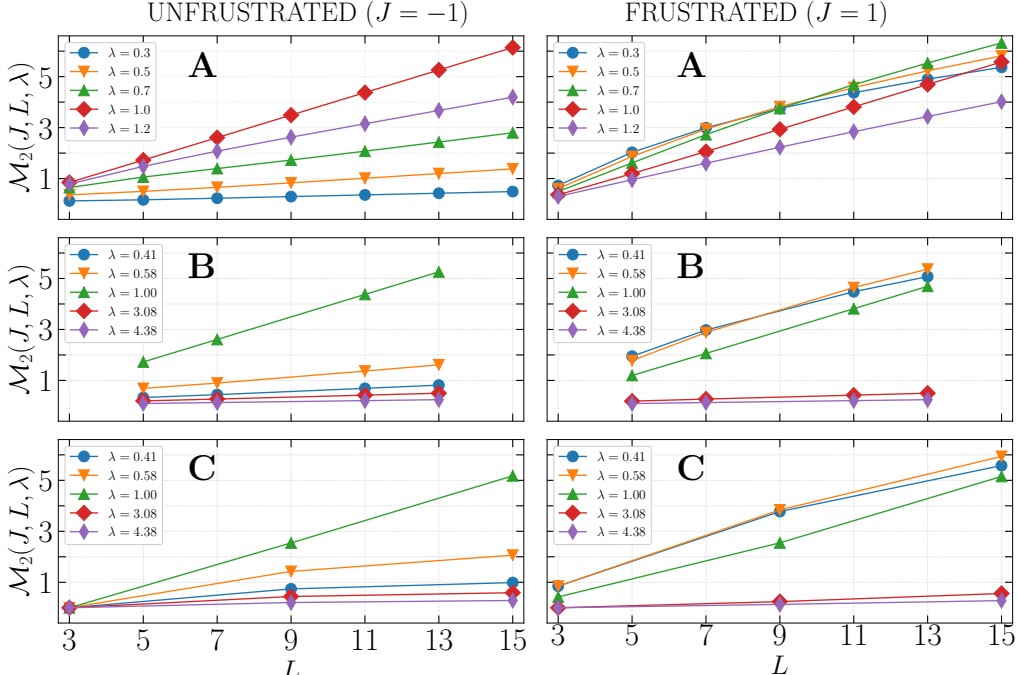

Figure 3: Comparison between the SRE for the unfrustrated system (left column) and the topologically frustrated (right column) for the different models: TFIM (panels A); CIM with $L$ not an integer multiple of 3 (panels B); CIM with $L$ an integer multiple of 3 (panels C). The different values of $\lambda$ are in the legends. While SRE always grows linearly with the chain length for the unfrustrated cases, topological frustration adds non-linear contributions that, in the thermodynamic limit, reduce to eq. (7).

In the thermodynamic limit, in the topologically frustrated phases, the SRE can be seen as the sum of a local term equal to the one of the corresponding unfrustrated model, and a second contribution coming from the delocalized $|W_k\rangle$ state. This means that, at least for large $L$, we expect that, in the frustrated phase,

$$\mathcal{M}_2(J=1, L, \lambda) = \mathcal{M}_2(J=-1, L, \lambda) + \mathcal{M}_2^W(L), \tag{7}$$

where $\mathcal{M}_2^W(L)$ is the SRE of a $W$-state that is given in (3). On the other hand, $\mathcal{M}_2(J=1, L, \lambda)$ stands for the SRE for the frustrated model while $\mathcal{M}_2(J=-1, L, \lambda)$ stands for the SRE evaluated in the unfrustrated one, obtained changing the sign of $J$ but keeping fixed the values of both $L$ and $\lambda$. On the contrary, immediately outside such a phase, the effect of frustration tends to disappear as $L$ increases.

To highlight this picture, in Fig. 4 we plot the behavior of the *relative frustrated SRE correction* that is the difference between the SRE for the frustrated and the unfrustrated models, for fixed values of $\lambda$ and $L$, normalized with $\mathcal{M}_2^W(L)$, i.e.

$$\mathcal{R}(L, \lambda) = \frac{\mathcal{M}_2(J=1, L, \lambda) - \mathcal{M}_2(J=-1, L, \lambda)}{\mathcal{M}_2^W(L)}. \tag{8}$$

In all the cases analyzed in the figure, the results are consistent with our picture. In fact, for $\lambda < 1$, i.e. when the system is in the frustrated phase, the quantity $\mathcal{R}(L, \lambda)$ tends asymptotically to 1 for large chains. In all other cases, it tends to vanish with $L$. It is worth noting that the values of the ratio are not exactly 1 or 0 but tend to these thresholds only for large $L$.

In the thermodynamic limit, quantities defined on finite supports converge to the same value for a frustrated system and its unfrustrated counterpart, but when the size is finite

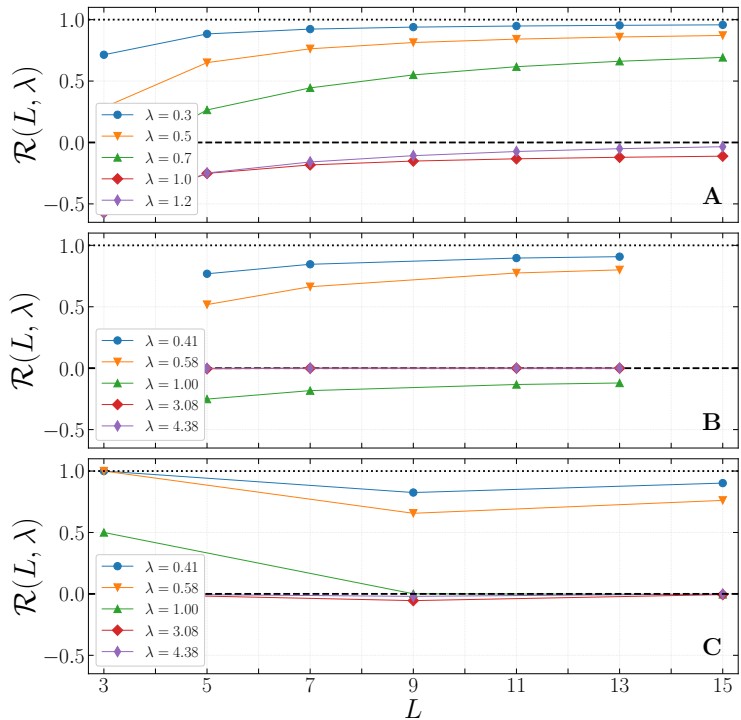

Figure 4: Behavior of the relative frustrated SRE correction $\mathcal{R}(L,\lambda)$ as function of $L$ for different models and values of $\lambda$: TFIM (panels A); CIM with $L$ not an integer multiple of 3 (panels B); CIM with $L$ an integer multiple of 3 (panels C).

there are corrections scaling like $1/L$ [51]. We stress that these corrections cannot explain the logarithmic dependence of the SRE on the size of the chain, since it provides only a size-independent correction to the extensive SRE of the frustrated model. Nevertheless, it allows quantifying the finite-size effects on the magic. In fact, in Ref. [42] the authors proved the local nature of the SRE for the unfrustrated Ising model by showing how its behavior is well mimicked by the SRE of the reduced density matrix of a single spin. In other words, labeled $\alpha_1(\lambda)$ the SRE of the reduced density matrix of a single qubit

$$\alpha_1(\lambda) = \log_2\left(\frac{1+m_z^2}{1+m_z^4}\right), \tag{9}$$

where $m_z = \langle\sigma_j^z\rangle$ is the expectation value of $\sigma_j^z$ over the ground state of the unfrustrated model, we have that the SRE $\mathcal{M}_2(J=-1,L,\lambda\neq1) \simeq L\alpha_1(\lambda)$. In the frustrated case, the expectation value of $\sigma_j^z$ acquires a correction proportional to $L^{-1}$ ($\langle\sigma_j^z\rangle = m_z + 2/L$), which, when plugged into (9), gives

$$\mathcal{M}_2(J=1,L,\lambda\neq1) \simeq L\alpha_1(\lambda) + 4m_z\left(\frac{1}{1+m_z^2} - \frac{2m_z^2}{1+m_z^4}\right). \tag{10}$$

We see that even accounting for the correction, a local approximation for the SRE fails to reproduce the non-local logarithmic growth coming from the $W$-states. This feature can also be connected to MPS representations. States that can be represented as translational invariant MPSs, such as the ground state of the unfrustrated Ising model, have extensive SRE [39]. In contrast, the $W$-state has no efficient representation as a translational invariant MPS [13,66], thus a violation of the extensive property of SRE is possible. Obviously, MPS representations of $W$-states with boundary tensors exist but, they are not translational invariant MPS. On the

other hand, one can find a translational invariant MPS representation for the $W$-states but with linearly increasing bond dimensions [67].

Furthermore, we note that (10) is not valid for the CIM since $m_z = 0$. This fact implies that in order to mimic the SRE for the unfrustrated model, we cannot limit the reduced density matrix of a single spin, but we have to take larger partitions, where we can recover similar behaviors.

# 5 Conclusions

In conclusion, in our work, we have considered a well-known family of states in the field of quantum information, namely the $W$-states. For this family, we have calculated the value of the SRE, and we have highlighted the logarithmic dependence of the latter on the size of the system. This non-zero SRE does not come from the individual states whose combination gives life to the $W$-states, but from the particular, non-local, superposition of them, which produces a completely delocalized SRE. In fact, the $W$-states are written in terms of states that are eigenstates of Pauli strings and therefore, taken individually, they have a null non-stabilization value.

We have shown how these states are realized naturally in a quantum spin chain close to its classical point when the system is topologically frustrated. The quantum term competing with the Ising interaction, but preserving the parity of the magnetization in a transverse direction selects a superposition of factorizable kink states which, through a Clifford circuit, can be exactly mapped into the $W$-states. Since any measure of non-stabilization remains unaffected by the application of a Clifford circuit, the ground state of the (nearly classical) spin chain and the $W$-state share the same, non-vanishing SRE which grows logarithmically with the number of qubits/spins. This behavior should be contrasted to that of non-frustrated systems, which are close to their classical point approach GHZ-state and have zero SRE.

Moving away from the classic point, the competition between the quantum interactions gives rise to an additional contribution that scales linearly with $L$ in a way that is analogous to that of the unfrustrated counterparts. Therefore, in a topologically frustrated system, the SRE is the result of the coexistence of a dominant local contribution and a subdominant one originating from the delocalized nature of the $W$-states.

This work combines quantities and concepts from two different fields: quantum computing and the theory of complex quantum systems, and provides new insights for both. From the many-body point of view, it offers further evidence of the difference between the same model with and without topological frustration and shows that the former has a much richer (non-local) complexity.

From a quantum information theory perspective, the results presented in this paper provide a new embedding of $W$-states in a physically realizable setting and a generalization of these states to a finite correlation length. Furthermore, the additional SRE of the frustrated systems, particularly relevant for microscopic and mesoscopic systems, can be a valuable resource for the design of devices based on topological frustrated models. To provide an example, let us consider fault-tolerant quantum computers, which are quantum computers with a physical error rate below a certain threshold. Accordingly with the threshold theorem [68–70], through quantum error correction schemes, the associated logical error can be suppressed to arbitrarily low levels. Fault-tolerant quantum computations are commonly run by state synthesis protocols, where a resource state $|\phi\rangle_{\text{ini}}$ combined with error-corrected Clifford operations is transformed into a target state $|\phi\rangle_{\text{target}}$ [71]. Magic (SRE) quantifies the non-stabilizer resources needed to synthesize a particular state or unitary [31, 72, 73]. As the SRE is invariant under Clifford unitaries, a necessary condition for state synthesis is that

$\mathcal{M}_2(|\phi\rangle_{\text{target}}) \le \mathcal{M}_2(|\phi\rangle_{\text{ini}})$. Thus, it bounds the minimum amount of magic resource states needed to simulate a state on a quantum computer. For example, the commonly used magic state $|T\rangle$ defined as $|T\rangle = \frac{1}{\sqrt{2}}(|0\rangle + e^{-i\pi/4}|1\rangle)$, which can be used to realize a single **T**-gate, that together with the Hadamard and the CNOT gates give access to universal quantum computation [74] has magic equal to $\mathcal{M}_2(|T\rangle) = \log_2(4/3)$. But, even if we consider $L = 3$, which represents the minimum size for which we can distinguish among $W$ and GHZ states, the SRE of a $W$-state is larger by $\log_2(4/3)$. Hence, a single three-qubit $W$-state, and therefore any topological frustrated one-dimensional system, could provide an amount of non-stabilizer resources sufficient for the realization of a **T**-gate.

## Acknowledgements

TH thanks L. Piroli for the discussions.

**Funding information**  JO recognizes the support of the Croatian Science Foundation under grant number HRZZ-UIP-2020-02-4559. AH acknowledges support from NSF award number 2014000. SMG, FF, and GT acknowledge support from the QuantiXLie Center of Excellence, a project co-financed by the Croatian Government and the European Union through the European Regional Development Fund – the Competitiveness and Cohesion (Grant KK.01.1.1.01.0004). FF and SMG also acknowledge support from the Croatian Science Foundation (HrZZ) Projects No. IP–2019–4–3321.

## A  Analytic derivations of SRE

In this Appendix, we detail the analytical derivations of the SRE for a $W$ state. Let us start by recalling the definition of the $W$ state in the $x$-basis (see (2)) as

$$|W\rangle = \frac{1}{\sqrt{L}} \sum_{i=1}^{L} \sigma_i^z |-\rangle^{\otimes L} . \tag{A.1}$$

Denoting with $P$ a generic Pauli string operator defined on the system of $L$ qubit, to determine the SRE given in (1), we have to evaluate the expectation value of the $W$ state over $P$, i.e.

$$\langle W| P |W\rangle = \frac{1}{L} \sum_{i,j=1}^{L} \langle -|^{\otimes L} \sigma_j^z P \sigma_i^z |-\rangle^{\otimes L} . \tag{A.2}$$

Let us consider separately the two different cases, $i = j$ and $i \ne j$, that will provide two different kinds of contribution to the SRE, i.e.

$$\mathcal{M}_2^W(L) \equiv \mathcal{M}_2(|W\rangle) = -\log_2 \left[ \frac{1}{2^L} \left( O_{i=j} + O_{i\ne j} \right) \right] . \tag{A.3}$$

When $i = j$, we have that only the string in which there is no $\sigma_k^y$ either $\sigma_k^z$ give a non-zero expectation value. Hence, the non-vanishing contribution will arrive only by $2^L$ strings that can be put in the form $P' = \bigotimes_{k=1}^{L} \sigma_k^\alpha$, where $\alpha \in \{0, x\}$. The absolute value of each term defined in eq. (A.2) depends on the number $l = 0, \dots, L$ of $\sigma_k^x$ operators in the string $P'$ and it is equal to $\|\frac{L-2l}{L}\|$. Taking into account all the possible combinations, the contributions of these terms is

$$O_{i=j} = \sum_{l=0}^{L} \left( \frac{L-2l}{L} \right)^4 \frac{L!}{l!(L-l)!} . \tag{A.4}$$

On the opposite, in the case $i \neq j$, the terms that provide a non-zero contribution lives in a different set. In fact, in this second case, only string operators of the form $P'' = \bigotimes_{k=1, k \neq i,j}^{L} \sigma_k^{\alpha} \otimes (\sigma_i^{\beta} \sigma_j^{\beta})$, where $\alpha \in \{0, x\}$ while $\beta \in \{y, z\}$ provide a non-vanishing contribution which absolute value is equal to $\frac{2}{L}$. Since all of them provide the same contribution to the magic, it is easy to see that $O_{i \neq j}$ can be written as

$$O_{i \neq j} = \sum_{l=0}^{L-2} 2 \left(\frac{2}{L}\right)^4 \frac{L(L-1)}{2} \frac{(L-2)!}{l!(L-2-l)!} \,. \tag{A.5}$$

Introducing both (A.4) and (A.5) in (A.3), and after a few simplifications we obtain

$$\mathcal{M}_2^W(L) = 3 \log_2(L) - \log_2(7L - 6) \,. \tag{A.6}$$

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
