# Peer review of "Complexity of frustration: a new source of non-local non-stabilizerness"

_SciPost Physics, doi:SciPost Phys. 15, 131 (2023)_

## Round 1 · Referee Report · Anonymous (Referee 1) · 2023-5-30

Report

The authors have made suitable modifications to their manuscript following the first round of review. I recommend publication in its current form.

---

## Round 1 · Referee Report · Anonymous (Referee 2) · 2023-7-26

Report

The Authors consider the finite-size scaling of the degree of “non-stabilizerness” of many-body systems, a quantity that in quantum computation determines the simulability with Clifford resources only. The analysis revolves around the case of Ising-like spin chains which exhibit a (linearly) extensive degeneracy at their classical point, due to some sort of (topological) frustration, i.e., one ferromagnetic defect in an otherwise anti-ferromagnetic background. Once an additional field is added (e.g., the common transverse magnetic field, but not only), the degeneracy is lifted and the ground-state becomes a translationally-invariant superposition of the classical ones. These can be interpreted as W-states, and some of the conclusions seem to hint at the fact that this constitutes a novel recipe to realize useful resource states for quantum computation purposes.

Per se, the topic is interesting and timely, and some of the results potentially useful across the fields of quantum computation/simulation and condensed matter physics. There are however some points to be clarified and some issues with the presentation to be solved, before publication on SciPost.

———

0) The nomenclature of “stabilizers”, “clifford gates”, and so on, is given for granted and not even briefly recalled, which could be desirable for self-containedness of the paper. The same applies to the paragraphs in the Conclusions where the Authors suddenly discuss about resources for fault-tolerant quantum computation, T-gates and so on, without having put the things in too much context / having recalled the basics.

1) In Fig. 2, the unfrustrated curves clearly display a peak (most probably even diverging) at the quantum critical point, while the unfrustrated curves do not: why exactly is that the case? From the analysis presented by the Authors in Figs. 3 & 4, and the main text, I was not able to reconstruct it, unless I overlooked something. By the way, why are panels B & C at such lower resolution (# points) than panel A? Any technical reason worth to be mentioned?

2) On page 10, end of Sec. 4, I do not understand the statement “In contrast, the W-state has no efficient representation as a translational invariant MPS [13]”, since it is rather easy to write down a MPS with bond-dimension 2, namely (to use the notation of Eq. (2)): $A_{0,0} = \left|-\right\rangle, \quad A_{0,1} = \sigma_z \left|-\right\rangle / \sqrt{L}, \quad A_{1,1} = \left|-\right\rangle$ and boundary vectors that force to start with index 0 and terminate with index 1. Since it seems that the Authors give a lot of meaning to this “non-representability”, could they clarify what they exactly mean? Does my objection compromise their conclusions?

3) While most of the text and the plots are focussed on the scaling of the different entropic quantities (in primis the “non-stabilizerness”) with the size of the system, the last paragraphs of the Conclusions seem instead to highlight that the size does not really matter for the usefulness of a state for quantum computational purposes, stating explicitly that “a single three qubit W-state, and therefore any topological frustrated one-dimensional system, could provide an amount of non-stabilizer resources sufficient for the realization of a T-gate.” Are these two aspects complementary, or is there one that should be regarded more as the element of novelty and core message of the work?

4) One even more general question, maybe an outlook for future studies, is how much of the picture outlined by the Authors in this work generalizes to other kind of spin chains (e.g., with symmetries different/larger than Z_2), and — more intriguingly — to spatial dimension larger than one, where frustrated classical Ising models may exhibit exponential degeneracy of the ground-state manifold, and quantum terms lift it by the mechanism called “order-by-disorder”… Can the Author provide any comment/outlook here?

———

Minor:

i) at page 4, around Eq.(2) it should read “W-state” or the verbs should be in their plural form…

ii) same page, footnote 1: N should be L, right?

iii) same footnote: formulated like this, the statement is true only for states of the computational basis and the Pauli matrix is \sigma_z (or anyway the one that defines the computational basis) — I guess the Authors mean instead that, if the state is factorized, one can always locally rotate the basis site-by-site, etc.

iv) page 5: “an invariant under spatial translation Hamiltonian” —> “a translationally-invariant Hamiltonian”

v) why does the quantity $\mathcal{R}(L, \lambda)$, which plays a central role in the analysis of the results, does not carry a name?

---

## Round 1 · Author Response

Warnings issued while processing user-supplied markup:

  • Inconsistency: plain/Markdown and reStructuredText syntaxes are mixed. Markdown will be used.
    Add "#coerce:reST" or "#coerce:plain" as the first line of your text to force reStructuredText or no markup.
    You may also contact the helpdesk if the formatting is incorrect and you are unable to edit your text.

Dear Editor,

We thank the referees for her/his positive judgment of our work and his/her suggestions and requests that have helped us to prepare a clearer version of the text. We are now confident that this revised version of our work will match all the requirements to be published in a journal as relevant as SciPost. Together with the changes suggested by the referee, we have also modified the format of our paper, adding a table of contents and the DOI of all the cited works to better adhere to the style of the journal. In the following, you can find a detailed reply to the referee's remarks

Referee -> The authors consider the relation between frustration in finite size many-body quantum systems and the degree of "non-stabilizerness", a concept which emerges from the framework of quantum computation and is related to the simulability of a system with classical devices using only Clifford resources. The authors consider a broad class of Hamiltonians which exhibit an extensive degeneracy at their classical point in parameter space (in this case, the authors essentially focus on an Ising model). By considering odd, finite size systems and periodic boundary conditions, the authors discuss that when a symmetry breaking term is added the ground states of these models are effectively W-type states. The implications of this are then considered for two exemplary cases: the quantum Ising model and the cluster-Ising model. The article is well written, and I think a relevant addition to the literature, and I feel merits publication in SciPost. While the analysis itself is relatively straightforward and deals with moderate size systems, I believe the impact of the present submission comes from the connections draw between disparate communities. There are a few points I would invite the authors to consider:

Authors -> We thank the referee for her/his positive judgment on our paper.

  1. In Fig 2 the authors show the behavior of the SRE as a function of system size for various fixed values of \lambda. Does the SRE reveal features of the underlying critical point? Looking at the behavior of the plots, it seems that the curves are indeed non-monotonic as a function of \lambda. I would be interested in seeing the behavior of \mathcal{M}_2 as a function of lambda for a moderate value of L. Indeed, we see from the lower right panel of Fig 2 that there is a sizeable gap between values of SRE observed depending on the phase the model is in.

Authors -> We agree with the referee that showing the behavior of the SRE as function of \lambda at fixed size of the chain can be a relevant piece of information that allows to better highlight the differences between frustrated and unfrustrated models. Hence, in the new version, we have added it for all the models considered in the paper, for dimensions of the systems up to L=15. In the new Fig.2 we see a clear difference between frustrated and unfrustrated models. In the last ones, the critical point at $\lambda=1$ is always characterized by a local maximum of the SRE. This peak is associated with the non-local behavior of magic shown in ref[42]. On the other hand, the frustrated model does feature such delocalization of magic also in the gapped phase (due to frustration).

Referee -> 2. The models the authors considered host a 2nd order QPT. Does the nature of the phase transition play an important role in the results?

Authors -> We thank the referee for her/his very relevant questions. To be honest, it is very hard to provide a precise answer to it. As we can see from figure 2 that the unfrustrated models shows a peak in correspondence of the QPT that are connected to the divergence of the correlation length that is a standard phenomenology of the second order QPT. Usually such a divergence is absent in the first order QPT, and hence we can expect that the local maximum of the SRE that we see in the new Fig.2 will be absent in a similar plot for a system showing a first order QPT. But this is very speculative and, at this moment, we cannot provide a general behavior. Overall, the point raised by the referee is of extreme interest, and we wish to focus on it in a future analysis.

Referee -> 3. There is a curious omission of the results for L=9 for the cluster-Ising model. Considering the small system sizes the authors are exploring and the conclusions being drawn based on them, I feel the including of this data point is important.

Authors -> For both the frustrated and the unfrustrated versions of the Cluster-Ising model, the systems with L integer multiple of three are very peculiar. In the second case, this difference was underlined in the seminal paper of Smacchia (ref. [62] of the paper), while in the frustrated counterpart, we have that the ground state of the system shows a three-fold degeneracy, which is completely absent for other odd values of $L$. For these reasons, in the first version of the paper we have decided to focus on the case with $L$ not an integer multiple of 3. Unfortunately, for our mistake, in the submitted version of the paper, the paragraph in which we explain this point was canceled. However, accepting the suggestion of the referee, we have decided to add to the new version also the case with the CIM with L an integer multiple of 3. But to underline its peculiar behavior, we have decided to keep the discussion about the SRE in the CIM separated in the two cases, i.e. L equal or different from an odd integer multiple of 3. All the plots were redrawn to take into account this change.

Referee -> Some minor points for the authors to consider: 1. In the abstract I believe there are a few grammatical errors: "grows logarithmic with" -> "grows logarithmically with" and "which the classic point belongs" -> "which the classical point belongs"

Authors -> We thank the referee for her/his carefully reading our paper. We have fixed these two points in the abstract.

Referee -> 2. Below Eq. 1 the set for P_j includes \sigma_k^x. Should this be \sigma_j^x?

Authors -> We thank again the referee for her/his carefully reading our paper. We have fixed the notation in the text.

Referee -> 3. Above Eq. 5 the authors explain the degeneracy of the models. I find the use of parentheses to caveat the sentence jarring and difficult to follow, i.e. "the CIM is equal to 1 (3) if and only if L is odd and is not (is) and interger...." I feel this can simply be written out explicitly and will be clearer to follow. Similarly, around equation 7 the use of parentheses could be alleviated and things written out explicitly for clarity.

Authors -> We agree with the referee that the two parts in the text were not clear. We have re-written the two parts making them clearer and easier to read.

Best regards To behalf of the authors Salvatore Marco Giampaolo

---

## Round 1 · List of Changes

Here is the list of changes we made to respond to the referee's remarks

  • We add a plot (fig 2 in the new version), and the relative discussion, showing the behavior of the SRE as function of \lambda at fixed size of the chain for all models analyzed;

  • We modified fig 3 (fig 2 in the old version) separating the CIM in two subcases: L integer multiple of 3 and L non integer multiple of 3;

  • We modified fig 4 (fig 3 in the old version) separating the CIM in two subcases: L integer multiple of 3 and L non integer multiple of 3;

  • We added a table of contents at the beginning of the paper

  • We added the DOI of all the cited papers

  • We made minor text corrections

All the changes in the paper are in blue.

---

## Round 2 · Referee Report · Anonymous (Referee 2) · 2023-8-8

Report

All previous points have been taken into consideration, and the manuscript has been noticeably improved with respect to all of them. It should be published without further revisions.

---

## Round 2 · Author Response

Warnings issued while processing user-supplied markup:

  • Inconsistency: Markdown and reStructuredText syntaxes are mixed. Markdown will be used.
    Add "#coerce:reST" or "#coerce:plain" as the first line of your text to force reStructuredText or no markup.
    You may also contact the helpdesk if the formatting is incorrect and you are unable to edit your text.

Dear editor

We have carefully read the report of the second referee. First, let us thank her/him for her/his positive evaluation of our work and for her/his suggestions which allowed us to further improve our paper. In the following you can find a detailed answer to all the points she/he raised

The Authors consider the finite-size scaling of the degree of “non-stabilizerness” of many-body systems, a quantity that in quantum computation determines the simulability with Clifford resources only. The analysis revolves around the case of Ising-like spin chains which exhibit a (linearly) extensive degeneracy at their classical point, due to some sort of (topological) frustration, i.e., one ferromagnetic defect in an otherwise anti-ferromagnetic background. Once an additional field is added (e.g., the common transverse magnetic field, but not only), the degeneracy is lifted and the ground-state becomes a translationally-invariant superposition of the classical ones. These can be interpreted as W-states, and some of the conclusions seem to hint at the fact that this constitutes a novel recipe to realize useful resource states for quantum computation purposes. Per se, the topic is interesting and timely, and some of the results potentially useful across the fields of quantum computation/simulation and condensed matter physics. There are however some points to be clarified and some issues with the presentation to be solved, before publication on SciPost.

We thank the referee for reading our work carefully and for her/his positive opinion.

0) The nomenclature of “stabilizers”, “clifford gates”, and so on, is given for granted and not even briefly recalled, which could be desirable for self-containedness of the paper. The same applies to the paragraphs in the Conclusions where the Authors suddenly discuss about resources for fault-tolerant quantum computation, T-gates and so on, without having put the things in too much context / having recalled the basics.

We agree with the referee that in some points we have taken for granted the reader's knowledge of some concepts of quantum information, and that this choice of ours may limit the number of researchers who approach the reading of our paper. Therefore, accepting her/his suggestion, both in the introduction and in the conclusions, we have added some sentences which, even without going into the details of the concepts introduced, can help to better understand their role in our work.

1) In Fig. 2, the unfrustrated curves clearly display a peak (most probably even diverging) at the quantum critical point, while the unfrustrated curves do not: why exactly is that the case? From the analysis presented by the Authors in Figs. 3 & 4, and the main text, I was not able to reconstruct it, unless I overlooked something. By the way, why are panels B & C at such lower resolution (# points) than panel A? Any technical reason worth to be mentioned? As the referee correctly observes, and as highlighted also in the article (page 7), the non-frustrated models show a maximum at the phase transition which becomes more and more evident as the size of the system increases. This phenomenon is consistent with the results shown in figure 3 where it is observed that the SRE for the non-frustrated models depends linearly on $L$ with a proportionality coefficient that increases as the critical point is approached. In agreement with the conclusions of the article in ref. [42] this behavior is related to the local nature of magic in non-frustrated systems. On the contrary, as highlighted in our work, the SRE for frustrated systems has two different origins.

As for the lower resolution of panels B and C, the reason is quite simple. Our first evaluations were made for the Ising model only. Then we decided to extend our analysis to the second model as well. Since the evaluation of the points take a long time, and the trend of the SRE of the Ising model seemed clear even with fewer points, we decided to reduce them. There is no other physical or numerical reason behind this choice. To prove this, while we were working on the text, we have evaluated other points and increased the resolution in the two panels.

2) On page 10, end of Sec. 4, I do not understand the statement “In contrast, the W-state has no efficient representation as a translational invariant MPS [13]”, since it is rather easy to write down a MPS with bond-dimension 2, namely (to use the notation of Eq. (2)): A_{0,0}=|−⟩,A_{0,1}=\sigma z|−⟩/\sqrt{L} A_{1,1}=|−⟩ and boundary vectors that force to start with index 0 and terminate with index 1. Since it seems that the Authors give a lot of meaning to this “non-representability”, could they clarify what they exactly mean? Does my objection compromise their conclusions?

As the referee correctly points out, there is a MPS representation of the W-state with boundary tensors, however this a non-translational invariant MPS. Translational invariant MPS representation with constant bond dimension do not exist (e.g. see arXiv:2306.16456 or arXiv:quant-ph/0608197). Note that one can find a translational invariant MPS for the W-state with linearly increasing bond dimension. The W-state (and any other state with logarithmically scaling stabilizer entropy) cannot have a translational invariant MPS representation (with constant bond dimension) as this requires an extensive stabilizer entropy. To make our point clearer, we have modified the text at the end of sec. 4.

3) While most of the text and the plots are focused on the scaling of the different entropic quantities (in primis the “non- stabilizerness”) with the size of the system, the last paragraphs of the Conclusions seem instead to highlight that the size does not really matter for the usefulness of a state for quantum computational purposes, stating explicitly that “a single three qubit W- state, and therefore any topological frustrated one-dimensional system, could provide an amount of non-stabilizer resources sufficient for the realization of a T-gate.” Are these two aspects complementary, or is there one that should be regarded more as the element of novelty and core message of the work?

As we have shown throughout our paper, there are many differences between the behavior of topologically frustrated and non-frustrated systems. The latter show a strictly local magic (except at the critical point) and a linear dependence of the SRE on the system size. Conversely, the magic for the former cannot be considered local and the dependence on the size of the system is not linear in the frustrated phase. This is the keystone of the article around which, as rightly observed by the referee, we have focused our analysis. However, in the conclusions, in order to further underline the difference between the two behaviors, the point indicated by the referee seemed useful to us. In fact, the difference between the two behaviors is even more evident near the critical point, where the SRE for the non-frustrated models vanishes, while the frustrated models show an excess of magic sufficient to simulate a T-gate.

4) One even more general question, maybe an outlook for future studies, is how much of the picture outlined by the Authors in this > work generalizes to other kind of spin chains (e.g., with symmetries different/larger than Z_2), and — more intriguingly — to spatial dimension larger than one, where frustrated classical Ising models may exhibit exponential degeneracy of the ground- state manifold, and quantum terms lift it by the mechanism called “order-by-disorder”… Can the Author provide any comment/outlook here?

We would like to thank the referee for this comment, which makes us understand that he/she read our paper with interest. However, before entering the merit of our reply, we need to clarify a point. The cluster-Ising model analyzed by us in the paper, has a more complex symmetry (Z_2 X Z_2, see Smacchia et al. in ref.[62]) and not the simple Z_2 of the Ising model. Coming back to the original question, there are several possible generalizations that we are currently working on. Among these, the generalization of the problem to 2D systems is one of the most interesting ones. The technical problem that we are facing right now, is that the size of the ground state manifold at the classical point grows a lot as the size of the system increases. But the first results, obtained so far on very small systems, are very promising.

Minor:

i) at page 4, around Eq.(2) it should read “W-state” or the verbs should be in their plural form…

We thank the referee for her/his careful reading. We have fixed the sentence

ii) same page, footnote 1: N should be L, right?

Yes

iii) same footnote: formulated like this, the statement is true only for states of the computational basis and the Pauli matrix is \sigma_z (or anyway the one that defines the computational basis) — I guess the Authors mean instead that, if the state is factorized, one can always locally rotate the basis site-by-site, etc.

Re-reading the part of our paper concerning points ii and iii raised by the referee, we noticed that the text was not very clear. We have rewritten it, removing the footnote.

iv) page 5: “an invariant under spatial translation Hamiltonian” —> “a translationally-invariant Hamiltonian”

We thank the referee for her/his suggestion that we have gladly accepted

v) why does the quantity R(L,λ) which plays a central role in the analysis of the results, does not carry a name?

Following the suggestion of the referee, we have named the quanity:" relative frustrated SRE correction"

---

## Round 2 · List of Changes

• We have changed fig.2 increasing the number of the points used in Panels B and C

  • In the first and last section of the paper, we have added some sentences to better introduce some concepts of quantum information and quantum computation.

  • At the end of section 4 we have changed the text to make the relation between MPS and W-state clearer.

  • We have added other 6 references.

  • Minor text corrections.

  • All changes made in the text are in blue

---

## Editorial Decision

published